# PAN/TiO$_2$ Ultrafiltration Membrane for Enhanced BSA Removal and Antifouling Performance

Yinshan Xie [1] , Xinning Wang [1], Hulin Li [1], Tao Wang [2], Wei Feng [2,*] and Jian Li [1,*]

1 Laboratory of Environmental Biotechnology, School of Environmental and Civil Engineering, Jiangnan University, Wuxi 214122, China; jameson618@163.com (Y.X.); sanni.wxn@gmail.com (X.W.); lihulin03@163.com (H.L.)
2 School of Food Science and Technology, Jiangnan University, Wuxi 214122, China; twang3813@gmail.com
* Correspondence: jjfengwei@jiangnan.edu.cn (W.F.); jian.li@jiangnan.edu.cn (J.L.)

**Abstract:** Membrane separation has been widely utilized to eliminate pollutants from wastewater. Among them, a polyacrylonitrile (PAN) ultrafiltration (UF) membrane has presented outstanding stability, and distinguished chemical and thermal properties. However, UF membranes inevitably incur fouling issues during their operation procedure caused by contaminant adhesion on the membrane surface, which would restrict the operational efficiency and increase the maintenance cost. The conventional physical and chemical cleaning is not an effective technique to reduce the fouling due to the additional chemical addition and inevitable structure damage. Recently, UF membranes combined with photocatalytic materials are suggested to be a useful approach to conquer the membrane fouling issues. Herein, TiO$_2$ nanoparticles were utilized to blend with a PAN casting solution for fabricating a composite UF membrane via a phase inversion method. With a certain TiO$_2$ addition, the obtained membranes presented an enhancement of hydrophilicity, which could promote the water permeability and antifouling performance. The optimized M3 membrane prepared with 15.0 wt% PAN and 0.6 wt% TiO$_2$ exhibited an excellent water permeability up to 207.0 L m$^{-2}$ h$^{-1}$ bar$^{-1}$ with an outstanding 99.0% BSA rejection and superior antifouling property. In addition, the photocatalytic TiO$_2$ nanoparticles endowed the M3 membrane with a remarkable self-cleaning ability under the UV irradiation. This facile construction method offered new insight to enhance the UF membrane separation performance with an enhanced antifouling ability.

**Keywords:** polyacrylonitrile; ultrafiltration; membrane fouling; TiO$_2$; photocatalysis





## 1. Introduction

Much attention has been paid to water quality and available water resources due to the speedy development of industrialization, populations and increased water pollution [1]. Various separation methods, e.g., ion exchange, membrane separation and adsorption, have been deployed to handle the pollutants from wastewater [2]. Membrane separation technologies, based on microfiltration (MF), ultrafiltration (UF) and nanofiltration (NF) membranes, have exhibited excellent advantages with high separation efficiency and low energy consumption, which outperform other traditional separation techniques [3,4]. Among them, UF membranes with high water permeation, a small footprint and modularity have been extensively deployed in water purification and wastewater treatment [5,6]. Generally, UF membranes are mainly fabricated with polymer materials such as polyethersulfone (PES), polyvinylidene fluoride (PVDF) and polyacrylonitrile (PAN) [7,8]. Compared to the other polymer materials, PAN possesses outstanding solubility in organic solvents, and a prepared PAN UF membrane exhibits distinguished chemical and thermal properties [9,10]. However, during wastewater treatment, especially for those systems containing organic compounds, the separated pollutants can be concentrated on the membrane surface and inside the pores via hydrophobic and electrostatic interactions. As a result, the gathered

contaminants need to be further removed or it could reduce the membrane separation efficiency and service life [11,12]. Hence, it remains challenging to explore an effective way for membrane fouling elimination.

Efficient methods such as adjusting membrane hydrophilicity, roughness and surface charge and optimizing the hydrodynamic condition have been utilized to improve the membrane fouling resistance [13,14]. Surface hydrophilicity majorization is an efficient method that contributes to reduce the membrane fouling. Thus, multiple approaches, such as surface grafting of antifouling materials, incorporating a hydrophilic substance during the membrane fabrication process and introducing inorganic particles into the polymer materials, have been employed to prepare a fouling-resistant UF membrane [15]. However, these modified UF membranes are still prone to sustain the fouling issues after a long-time filtration via the conventional physical and chemical cleaning, which is derived from the irreversible fouling [16]. Currently, advanced oxidation processes (AOPs), such as photocatalysis, Fenton and ozonation, have proven to be effective technologies to degrade the organic pollutants [17]. Among these AOPs, photocatalysis can effectively handle the contaminants with less chemical reagent usage and energy consumption [18,19]. Thus, filtration membranes combined with photocatalytic oxidation technology can be an efficient method to alleviate the membrane fouling issues [20,21]. Specifically, an electron of the semiconductor photocatalyst inside the membrane can be excited from the valence band (VB) to the conduction band (CB) facilitated with visible or UV light [22,23]. Thus, a positive charge and electron–hole pair are formed in the VB band during the process, which can generate the hydroxyl radicals ($\cdot$OH) and eventuate the degradation of contaminants on the membrane surface [24,25]. For example, a nano-ZnO particle was employed in the fabrication of a membrane matrix to obtain a photocatalysis PVDF UF membrane with a phase inversion approach, which exhibited enhanced fouling resistance and excellent self-cleaning under UV irradiation [26]. Meng et al. prepared a g-$C_3N_4$/$Bi_2MoO_6$-blended polysulfone UF membrane to improve the bovine serum albumin (BSA) rejection and antifouling performance with a high photocatalytic self-cleaning property under UV light irradiation [27]. A polysulfone (PSF) UF membrane combined with resorcinol–formaldehyde (RF) and a $\beta$-FeOOH nanoparticle exhibited an excellent water permeability ($376.3 \text{ L m}^{-2} \text{ h}^{-1} \text{ bar}^{-1}$) with BSA retention (97.5%), which could also effectively degrade the organic fouling under visible light illumination based on the cooperation of $H_2O_2$ [28].

Based on the highly photocatalytic, hydrophilic and inexpensive properties, titanium dioxide ($TiO_2$) has been broadly deployed in water treatment [29,30]. Previous studies have proved that membranes decorated with $TiO_2$ particles could improve the membrane separation performance, porosity and tensile strength [31]. With the desired hydrophilic properties, immobilization of $TiO_2$ particles endows the obtained membrane with enhanced hydrophilicity, inducing a better water permeability, and fouling resistance [32]. In addition, $TiO_2$ can be utilized to modify the membrane structure. A highly interconnected pore structure of a PVDF membrane was obtained with the introduction of $TiO_2$ into the matrix, resulting in a promoted pure water flux [33]. Moreover, immobilization of $TiO_2$ particles with UF membranes can effectively remove organic foulants in the presence of UV illumination. Accordingly, a $TiO_2$-based UF membrane could absorb incident photons and generate $\cdot$OH for decomposition of the contaminants existing on the membrane surface [34]. Damodar et al. fabricated composite PVDF membranes via mixing $TiO_2$ nanoparticles in a casting solution and the derived UF membrane presented a significantly improved antifouling property of the membrane under UV irradiation [35]. Thus, incorporation of $TiO_2$ particles into the UF membrane can improve the membrane antifouling performance and extend the membrane reusability.

In this work, we decorated a PAN UF membrane with $TiO_2$ particles through a phase inversion method for wastewater purification with improved membrane performance, including the pure water permeability, BSA rejection, flux recovery rate and self-cleaning property. Moreover, surface morphologies and chemical composition of the obtained membranes were explored. Meanwhile, the effects of PAN and $TiO_2$ particle concentration for

the membrane separation performance were detected. In addition, the self-cleaning ability of fabricated membranes and the mechanism were investigated under UV irradiation.

## 2. Results and Discussion

### 2.1. Morphologies of Membranes

The surface and cross-section morphologies of the obtained UF membranes are presented in Figure 1. According to Figure 1a, the M0 membrane possessed a microporous and smooth structure, which was generated with the phase inversion of the casting solution during the phase-conversion process [36]. With the introduction of $TiO_2$, crystal nuclei around 50 nm were uniformly exposed on the M1 membrane surface (Figure 1b). After an adequate concentration of $TiO_2$ was introduced to the membrane (Figure 1c,d), the M3 membrane presented a uniform distribution of particles with a small size on the membrane surface. However, continuously increased $TiO_2$ particle loading leads to the aggregation of the particles (Figure 1e,f), which may clog the membrane pores. Meanwhile, accumulation of $TiO_2$ particles would render the membrane surface rougher, resulting in a serious membrane fouling and a decline in water permeability. In addition, the PAN-based membranes displayed a typical asymmetric cross-sectional porous structure with finger-shaped macrovoids (Figure 1g–i), which was consistent with the non-solvent during the phase inversion process [37]. Meanwhile, the deposition of $TiO_2$ particles inside the membranes did not significantly change the cross-section of the PAN membrane, indicating an excellent accommodation of the PAN matrix and $TiO_2$ particles.

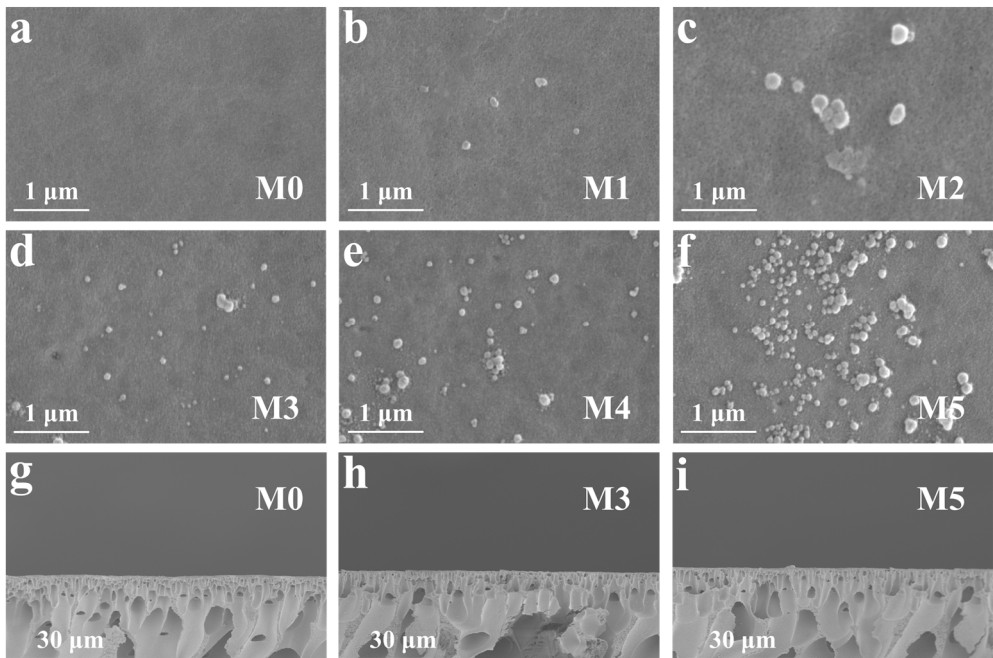

**Figure 1.** Surface SEM morphologies of (**a**–**f**) M0-M5 membranes; cross-sectional SEM morphologies of (**g**) M0, (**h**) M3 and (**i**) M5 membranes.

The XRD patterns of the obtained membranes are displayed in Figure 2. According to the XRD spectrum of M0, a broad diffraction peak was found around $2\theta = 17.0°$, corresponding to the (100) crystallographic plane of the PAN hexagonal structure [38,39]. As mentioned, the shape of the M0 membrane represented that the PAN membrane mainly possessed the amorphous structure with a negligible crystal feature [29]. Regarding the introduction of $TiO_2$, a new diffraction peak appeared at around $2\theta = 26.5°$ for M1–M5 membranes, which was attributed to the (101) crystal planes of $TiO_2$ nanoparticles [40], indicating the successful loading of $TiO_2$ particles on PAN UF membranes.

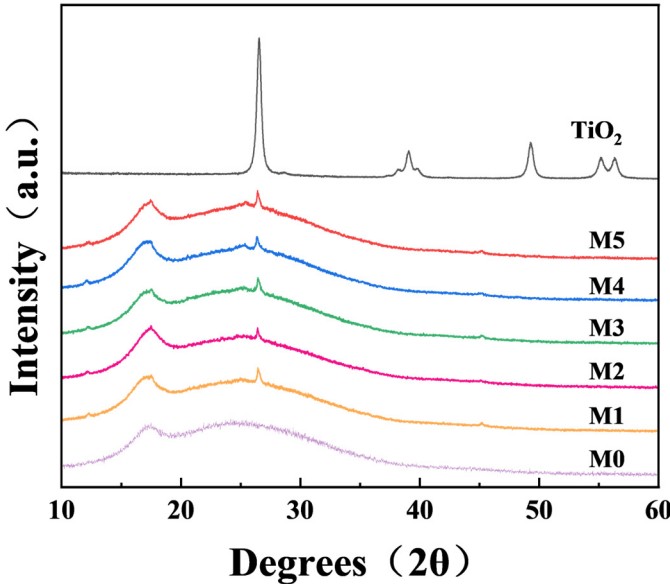

**Figure 2.** XRD spectra of prepared UF membranes.

## 2.2. Chemical Properties of the Membranes

FTIR was employed to detect the chemical compositions of the synthetic membrane surface (Figure 3). As exhibited in Figure 3a, four distinctive absorption peaks of PAN membranes at 2940 cm$^{-1}$, 2243 cm$^{-1}$, 1729 cm$^{-1}$ and 1453 cm$^{-1}$ corresponded to the stretching vibrations of C-H, -C ≡ N-, -C=O and -CH$_2$, respectively [41,42]. The peaks around 3500 cm$^{-1}$ were assigned to the bending vibration of -OH bonds of the TiO$_2$ particles [43]. In addition, the spectrum of TiO$_2$ particles exhibited a steep declined intensity of 1000 cm$^{-1}$ to 500 cm$^{-1}$, manifesting the existence of the O-Ti-O bond [29]. As a result, the addition of TiO$_2$ nanoparticles into the PAN membrane rendered an increased peak around 900 cm$^{-1}$ to 700 cm$^{-1}$ in Figure 3b, indicating the successful inclusion of TiO$_2$ nanoparticles into the PAN membrane matrix.

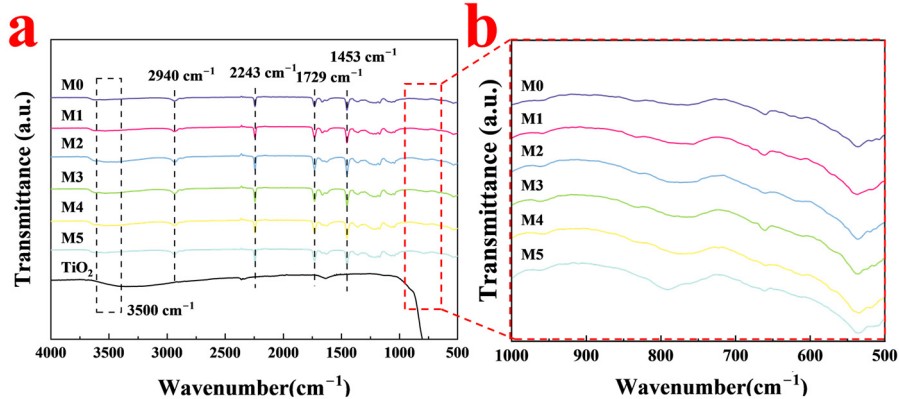

**Figure 3.** (**a**) Complete and (**b**) partial FTIR spectra of various membranes.

The hydrophilicity of the TiO$_2$-decorated membranes was revealed (Figure 4a) with a sessile drop technique. Compared to the M0 membrane, contact angles for M1 to M5 membranes declined, which was caused by the intrinsically hydrophilic TiO$_2$ particles improving the hydrophilicity of the membranes [44]. Meanwhile, the addition of TiO$_2$ nanoparticles improved the PAN membrane surface, which could promote the relevant wettability based on the Wenzel equation [45]. With an enhanced hydrophilicity, composite UF membranes could promote the water permeability and antifouling performance [46]. However, minor changes of the contact angle were detected for M4 and M5 membranes

compared with the M3 membrane. It was mainly attributed to the accumulation of $TiO_2$ particles on the M3 membrane, inducing a rougher surface with decreased hydrophilicity [37]. In addition, the incorporation of hydrophilic $TiO_2$ particles endowed the PAN polymer with a faster solvent and non-solvent exchange speed over the phase inversion of the casting solution [47]. Thus, a continuous improvement in membrane porosity was detected for the obtained M0 to M5 membranes (Figure 4b).

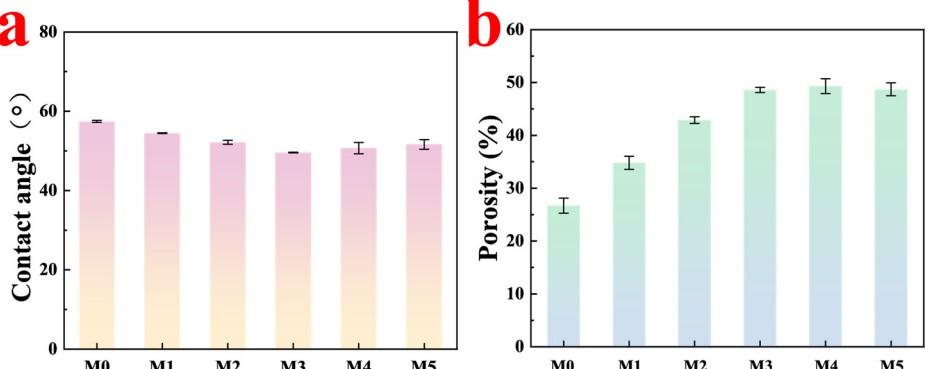

**Figure 4.** (**a**) Contact angle and (**b**) porosity results of prepared membranes.

### 2.3. Separation Performance of the Membranes

It was demonstrated that the structures and properties of the PAN/$TiO_2$ UF membranes were influenced using the concentrations of PAN power and $TiO_2$ loading, which could affect the resultant membrane performance. The pure water permeability and BSA rejection of obtained membranes are exhibited in Figure 5. As presented in Figure 5a, the water permeability of the $PAN_{13}$ membrane was measured up to 300.5 L $m^{-2}$ $h^{-1}$ $bar^{-1}$ with limited BSA rejection of 48.6%. With an increased viscosity of the PAN casting solution, a gradually declined water permeation and improved BSA rejection were detected, which caused the value of the $PAN_{15}$ membrane to reach 120.2 L $m^{-2}$ $h^{-1}$ $bar^{-1}$ and 99.1%. In addition, the water permeability suffered a decline with an increased PAN polymer concentration, while the BSA rejection remained almost constant. Considering the water permeability and BSA rejection, $PAN_{15}$ was regarded as the optimized membrane for fabricating the composite membranes with different additions of $TiO_2$ particles.

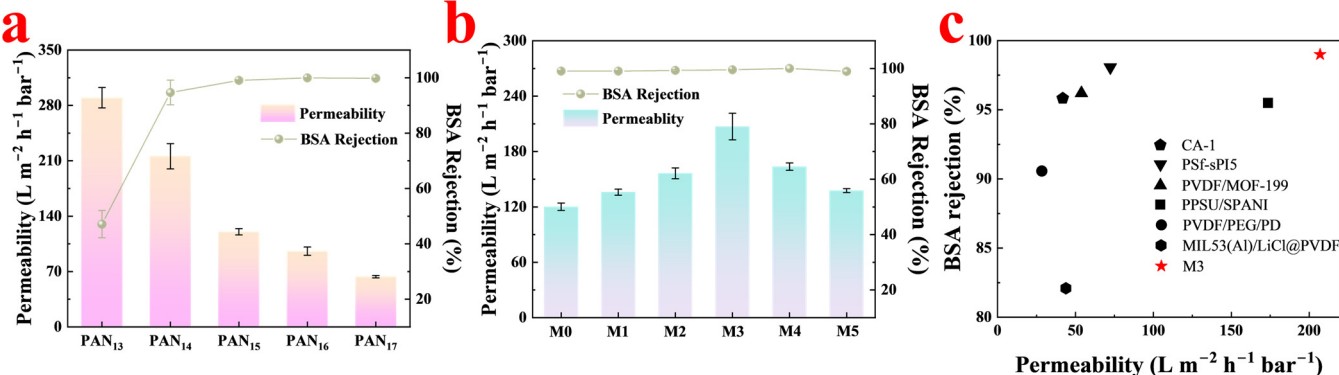

**Figure 5.** Water permeability and BSA rejection for (**a**) PAN; (**b**) PAN/$TiO_2$ membranes; (**c**) performance comparison of M3 membrane with other UF membranes.

As presented in Figure 5b, PAN UF membranes decorated with $TiO_2$ presented an improved water permeability from M0 to M3 (207.0 L $m^{-2}$ $h^{-1}$ $bar^{-1}$), which was explained with the higher hydrophilic membrane surface and porosity. However, the water permeability suffered a decline with the excess $TiO_2$ particle loading, which was consistent with the result of surface morphologies. The excess $TiO_2$ particles could gather on the

PAN membrane surface and blocked the pores of the membrane, inducing an enhanced resistance for transferring water molecules [32]. Furthermore, $TiO_2$ nanoparticles migrated into the PAN membrane during the phase inversion process and formed the agglomerates, increasing the membrane thickness and roughness, which led to a flux decline [48]. Simultaneously, the BSA rejection for the $TiO_2$-decorated PAN membranes remained above 99.0%. Figure 5c and Table 1 compare the separation performance of the M3 membrane with other reported UF membranes, indicating a superiority in water permeability and BSA rejection for the M3 membrane.

**Table 1.** Performance comparison of M3 membrane with other UF membranes.

| Membrane | Operating Pressure (bar) | Permeability ($L\ m^{-2}\ h^{-1}\ bar^{-1}$) | BSA Concentrations (g/L) | BSA Rejection (%) | Ref. |
|---|---|---|---|---|---|
| CA-1 | 3.5 | $41.6 \pm 0.4$ | 1.0 | $95.9 \pm 0.6$ | [49] |
| PSf-sPI5 | 5.0 | $72.1 \pm 0.8$ | 0.8 | $98.0 \pm 0.5$ | [50] |
| PVDF/MOF-199 | 3.5 | $54.2 \pm 0.4$ | 1.0 | $96.2 \pm 0.2$ | [51] |
| PPSU/SPANI | 1.5 | $173.3 \pm 2.6$ | 1.0 | $95.5 \pm 0.6$ | [52] |
| PVDF/PEG/PD | 4.1 | $28.8 \pm 0.6$ | 1.0 | $90.5 \pm 0.5$ | [53] |
| MIL53(Al)/LiCl@PVDF | 1.0 | $43.6 \pm 1.0$ | 1.0 | $82.1 \pm 0.9$ | [7] |
| M3 | 2.0 | $207.0 \pm 10.4$ | 1.0 | $99.0 \pm 0.3$ | This work |

However, the membrane would suffer from membrane fouling during the BSA filtration. The membrane antifouling properties were investigated through a filtration of the BSA solution (Figure 6). As shown in Figure 6a, the M0 membrane possessed the value of 16.30%, 90.22% and 83.71% for *FRR*, $R_t$ and $R_{ir}$, respectively. This result revealed that the M0 membrane suffered a serious membrane fouling formed with the adhesion of BSA molecules on the membrane surface. With the addition of $TiO_2$, the value of *FRR* for obtained membranes continuously improved to 37.13% for the M3 membrane. Meanwhile, the $R_t$ and $R_{ir}$ declined to 74.65% and 62.87%, respectively. The increased *FRR* and decreased $R_t$ and $R_{ir}$ value for the M3 membrane manifested the enhanced antifouling ability, which was attributed to the improved hydrophilicity of the membrane surface decorated with $TiO_2$ particles [54]. However, the excess $TiO_2$ addition endowed the M4 and M5 membranes with inferior *FRR* properties. This was caused by multiple-$TiO_2$-particle incorporation rendering a rougher membrane surface, resulting in enhanced adsorption to BSA molecules and a reduced membrane flux recovery ratio. Overall, by comprehensively considering the water permeability, BSA rejection and antifouling property, the M3 membrane was deemed as the optimized membrane.

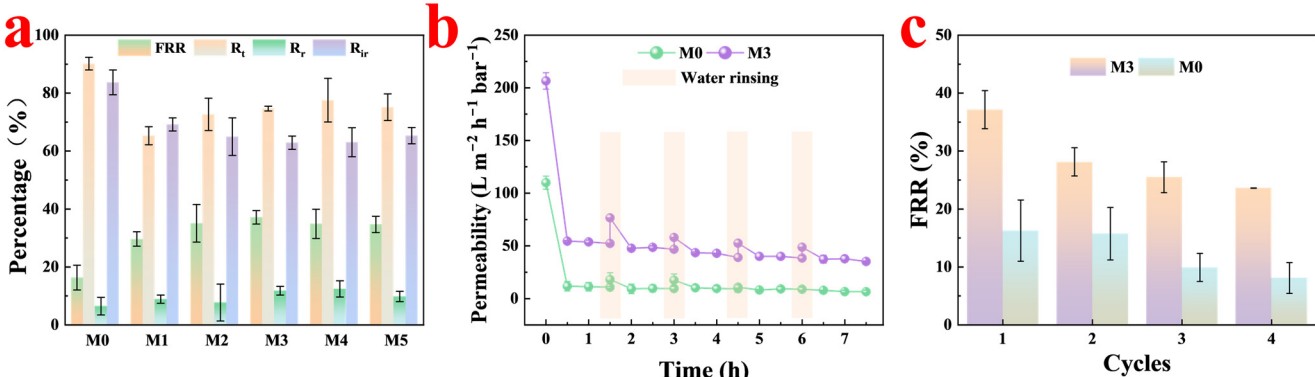

**Figure 6.** (**a**) The *FRR*, $R_t$, $R_r$ and $R_{ir}$ of membranes during the BSA ultrafiltration; (**b**) membrane performance and (**c**) *FRR* value in the process of water rinsing cycles.

A cyclic four-step ultrafiltration test for the BSA rejection solution was adopted to further measure the membrane antifouling performance in Figure 6b. It could be found that M0 and M3 membranes possessed stable BSA rejection above 99.0%, while for M0 and M3 membranes, the BSA fouling endowed the membranes with a continuous water permeability decline in the 1.5 h BSA filtration, which dropped to 10.8 and 52.3 $L\,m^{-2}\,h^{-1}\,bar^{-1}$, respectively. After that, the fouled membranes were washed in a pure water filtration, and obtained a certain flux recovery under the hydraulic shear force. Figure 6c summarizes the *FRR* of M0 and M3 membranes during the period. According to the data, the water permeability of the M0 and M3 membranes recovered to 18.07 and 76.55 $L\,m^{-2}\,h^{-1}\,bar^{-1}$ after the first water rinsing, which possessed the *FRR* of 16.27% and 37.13%, respectively. After that, the membranes were continuously filtrated in the BSA solution and water. It could be found that water permeability of the membranes dropped constantly in the BSA solution with the decreased *FRR* value, and that of the M3 membrane was higher than that of the M0 membrane in the process, while the water permeability had merely 23.58% *FRR* caused by the multiple contaminants adhered to the M3 membrane. Thus, the resultant M3 membrane incorporated with $TiO_2$ particles possessed a higher flux recovery rate compared to the pristine M0 membrane, but the extreme membrane fouling still existed after four BSA ultrafiltration tests.

### 2.4. Self-Cleaning Performance of the Membranes

For pursuing a better membrane antifouling performance, the designed membranes were tested in BSA ultrafiltration combined with UV illumination. Figure 7 presents the self-cleaning performance of the obtained membranes in a four-step filtration–irradiation process. Similarly, the water permeability of M0 and M3 membranes suffered a quick decrease in the 1.5 h BSA filtration as was discussed above (Figure 7a). After exposure to UV light irradiation for another 1.0 h, the M3 membrane decorated with photocatalytic $TiO_2$ particles derived an obvious permeation recovery, which reached 113.2 $L\,m^{-2}\,h^{-1}\,bar^{-1}$ with 52.41% *FRR* as depicted in Figure 7b. Compared with the value of water rinsing, the photocatalytic irradiation process endowed the M3 membrane with a great *FRR* promotion. However, the water permeability of the M0 membrane merely recovered to 20.2 $L\,m^{-2}\,h^{-1}\,bar^{-1}$, which was consistent with the value of water rinsing, revealing the useless effect of UV irradiation to the M0 membrane without $TiO_2$ particles. After that, the membranes were transferred to the same ultrafiltration process as before for four cycles. It demonstrated that the water permeability and *FRR* value of the M3 membrane finally declined to 63.85 $L\,m^{-2}\,h^{-1}\,bar^{-1}$ and 30.0% in the following cycles, respectively. It was explained with the not-fully-covered photocatalytic $TiO_2$ nanoparticles on the membrane surface, which could not degrade the whole of the BSA contaminants. Meanwhile, the M0 membrane also possessed a dropped water permeability and *FRR* value, which was consistent with Figure 6c. Thus, the M3 membrane cooperated with UV illumination and derived a great antifouling improvement compared to both the M0 membrane and M3 membrane without irradiation. In addition, M0 and M3 membranes maintained an excellent rejection above 99.0% towards BSA, indicating the high UV-irradiated stability of the PAN-based membrane. These results suggested that M3 incorporated with the $TiO_2$ membrane possessed an outstanding self-cleaning property and reusability.

The photoluminescence (PL) spectral was utilized to detect the photocatalytic property of the M3 membrane under the UV irradiation (Figure 8). As mentioned, terephthalic acid (TA) could be adopted to investigate the existence of ·OH radicals [55]. It could be found that a distinctive peak at λ = 425 nm was presented, which was associated with hydroxyterephthalic acid (HTA) [56]. With the addition of the M0 membrane, the corresponding solution displayed no difference with the pristine TA solution. However, a continual improvement in the fluorescence intensities was measured with the membranes incorporated with $TiO_2$ particles, indicating the production of ·OH radicals in the UV irradiation process. In addition, the increased $TiO_2$ particle concentration rendered the PL spectral with a continuous enhancement, which represented the generation of more ·OH

radicals. Thus, with the existence of ·OH radicals, the fouled M3 membrane could present a remarkable self-cleaning performance.

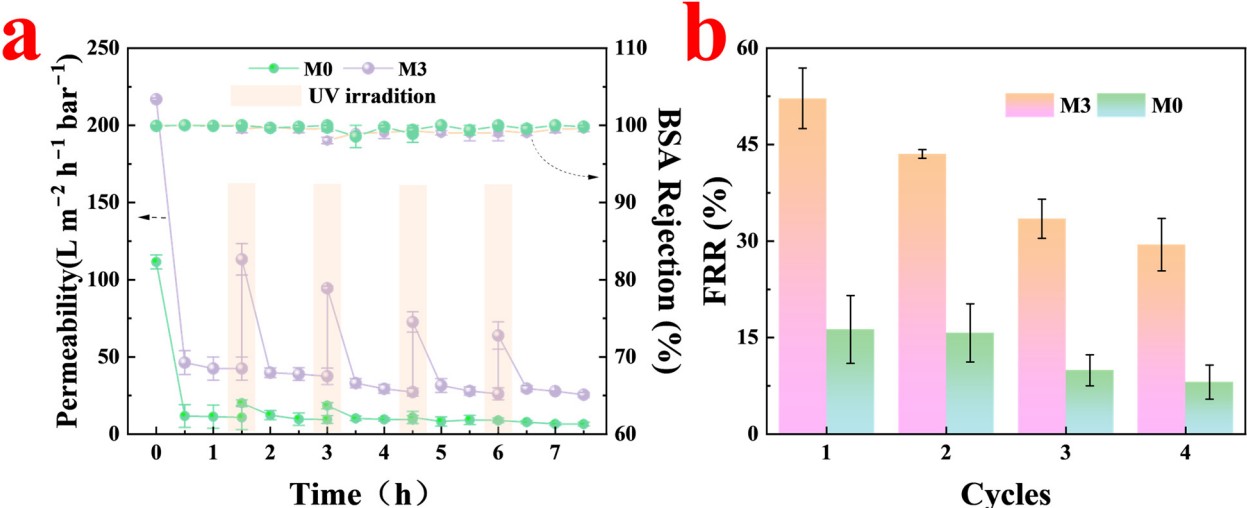

**Figure 7.** (**a**) Membrane performance and (**b**) *FRR* value in the process of UV irradiation cycles.

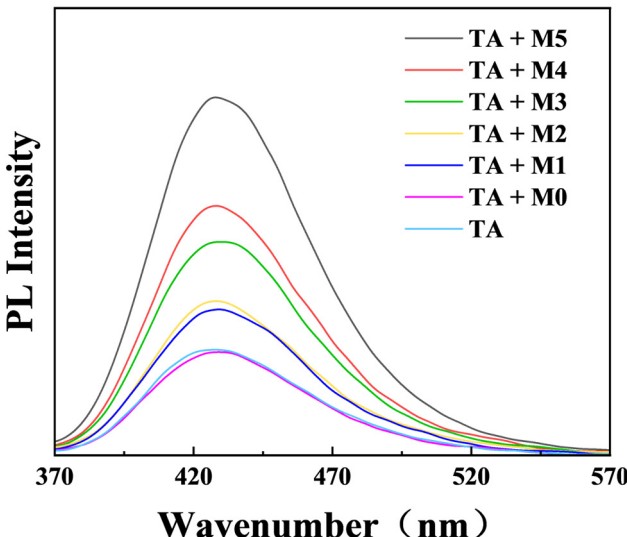

**Figure 8.** PL spectra of the solution including TA with different membranes.

## 3. Materials and Methods

### 3.1. Materials

N,N-Dimethylformamide (DMF, 99.7% purity) and terephthalic acid ($H_2$BDC, 99.0% purity) were provided by Sigma-Aldrich (Shanghai, China). Polyacrylonitrile (PAN) (MW = 100 kDa) was purchased from Solvey (Shanghai, China) Co., Ltd. Bovine serum albumin (BSA, MW = 66 kDa) and titanium dioxide ($TiO_2$, 99.7% purity) were supplied by Shanghai Aladdin Biochemical Technology Co., Ltd (Shanghai, China).

### 3.2. Preparation of PAN/$TiO_2$ Membranes

The fabrication process of PAN membranes is shown in Figure 9. Firstly, the PAN powders were dried in an oven for 12 h at 60 °C. Then, various qualities of PAN powders were dispersed in DMF (13 wt% to 17 wt%) and stirred for 12 h at 70 °C to obtain a homogeneous casting solution. After defoaming for 4 h, the prepared solution was poured onto a glass plate and scraped using a casting knife with the gap of 200 μm; then, the liquid

films were immersed into the deionized (DI) water and soaked for 24 h to obtain the PAN membranes.

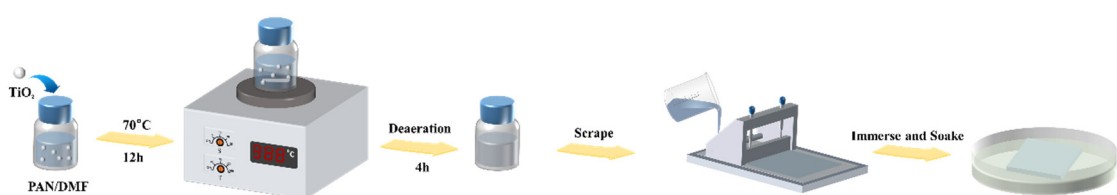

**Figure 9.** Schematic illustration of the preparation process for PAN/TiO$_2$ membranes.

For PAN/TiO$_2$ composite UF membranes, different amounts of TiO$_2$ powders were firstly dispersed in DMF and sonicated for 30 min. After that, a certain concentration of PAN powders was added into the prepared mixture solution and the above membrane fabrication operation was repeated to obtain the PAN/TiO$_2$ composite membranes. The corresponding composition of the membranes is listed in Table 2.

**Table 2.** Casting solution composition of the membranes.

| Membrane ID | Compositions | | |
|:---:|:---:|:---:|:---:|
| | DMF (mL) | PAN (wt%) | TiO$_2$ (wt%) |
| PAN$_{13}$ | 20.0 | 13.0 | - |
| PAN$_{14}$ | 20.0 | 14.0 | - |
| PAN$_{15}$/M0 | 20.0 | 15.0 | - |
| PAN$_{16}$ | 20.0 | 16.0 | - |
| PAN$_{17}$ | 20.0 | 17.0 | - |
| M1 | 20.0 | 15.0 | 0.2 |
| M2 | 20.0 | 15.0 | 0.4 |
| M3 | 20.0 | 15.0 | 0.6 |
| M4 | 20.0 | 15.0 | 0.8 |
| M5 | 20.0 | 15.0 | 1.0 |

*3.3. Membrane Characterization*

A scanning electron microscope (SEM, Hitachi SU8010, Japan) was deployed to inspect the morphologies of dried UF membranes. The chemical composition of the membranes was revealed with Fourier Transform Infrared (FTIR, Bruker Vertex 70, Switzerland). X-ray diffraction (XRD, Rigku Ultima IV, Japan) was used to investigate the crystal phase of the membrane. The hydrophilicity of the membranes was detected with a water contact angle device (OCA20, Dataphysics Instruments, Germany).

Membrane porosity was investigated with a reported approach [57]. Specifically, a cut membrane (2 cm × 2 cm) was immersed in DI water for 24 h; then, the excess water on the surface was carefully swiped and the weight of the obtained membrane was noted as $W_1$ (g). After that, the wet membrane was placed into an oven at 60 °C for 24 h and weighed as $W_2$ (g). The membrane porosity ($\varepsilon$) was measured as follows:

$$\varepsilon = \frac{\frac{W_1 - W_2}{D_w}}{\frac{W_1 - W_2}{D_w} + \frac{W_2}{D_p}} \times 100\% \tag{1}$$

where $D_w$ and $D_p$ are the water density (0.998 g/cm$^3$) and polymer density (1.25 g/cm$^3$), respectively.

### 3.4. Ultrafiltration and Antifouling Performance of Membranes

The separation performances of prepared UF membranes, concerning water permeability and BSA rejection, were evaluated with a crossflow cell setup (Hangzhou Saifei Membrane Separation Technology Co., Ltd., Hangzhou, China) with an effective area of 7.07 cm$^2$ as shown in Figure 10.

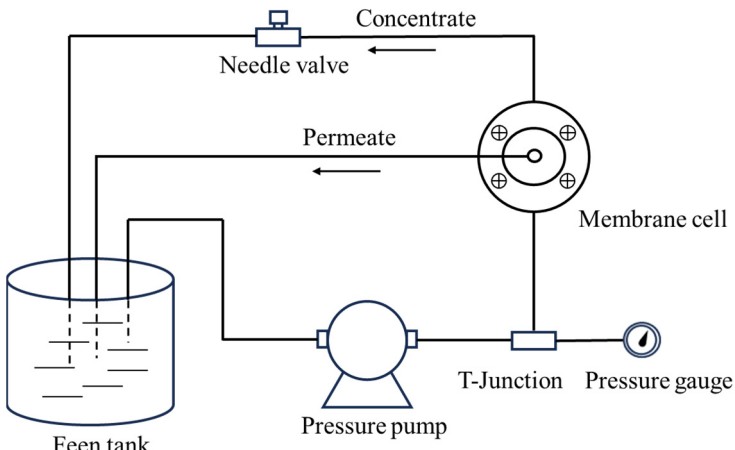

**Figure 10.** The schematic diagram of the crossflow cell setup.

After the pre-pressure process (0.2 MPa, 30 min), the pure water permeation ($J$) and BSA rejection ($R$) were calculated as follows:

$$J = \frac{V}{A \times t \times P} \tag{2}$$

$$R = \left(1 - \frac{c_p}{c_f}\right) \times 100\% \tag{3}$$

where $V$, $A$, $t$ and $P$ are permeated volume (L), filtration area (m$^2$), time (h) and operation pressure (bar). $c_p$ and $c_f$ represent the BSA concentration of the permeate and feed solutions, respectively. The feed BSA concentration was 1.0 g/L and a UV-vis absorption spectrometer (Shimadzu UV1800, Japan) was applied to measure the concentration of BSA.

The antifouling performance of membranes was detected with the flux recovery ratio ($FRR$), total fouling ($R_t$), the reversible fouling ratio ($R_r$) and the irreversible fouling ratio ($R_{ir}$) as follows [36,58]:

$$FRR = \frac{J_{w2}}{J_{w1}} \times 100\% \tag{4}$$

$$R_t = \left(1 - \frac{J_P}{J_{w1}}\right) \times 100\% \tag{5}$$

$$R_r = \left(\frac{J_{w2} - J_P}{J_{w1}}\right) \times 100\% \tag{6}$$

$$R_{ir} = \left(\frac{J_{w1} - J_{w2}}{J_{w1}}\right) \times 100\% = R_t - R_r \tag{7}$$

Specifically, the pure water permeability ($J_{w1}$) was detected at 0.2 MPa for 30 min. After that, a 1 g/L BSA solution was utilized as a feed solution and the permeability ($J_P$) was recorded every 10 min in a 30 min period. Then, the membranes were washed with pure water for 30 min and the water permeability ($J_{w2}$) of the refreshed membrane was obtained.

### 3.5. Self-Cleaning Performance of Membranes

The self-cleaning property of the PAN/TiO$_2$ membranes was tested with a photocatalytic degradation experiment of BSA using a 300 W xenon lamp (Perfect Light, PLS-SXE 300). Firstly, the membrane was pre-compacted in a 1 g/L BSA solution for 1.5 h at 2.0 bar, and the permeation and BSA rejections were recorded. After the filtration, a fouled membrane was transferred into 50 mL of DI water to be irradiated under UV light for 1 h. Meanwhile, a fluorescent probe test was deployed to investigate the mechanism during the self-cleaning process under UV irradiation, which was reported in the published literature [55].

### 4. Conclusions

In this work, a photocatalytic self-cleaning PAN UF composite membrane with TiO$_2$ decoration was synthesized via a facile phase inversion method. The characterization of membranes exhibited that the TiO$_2$ particles loaded on the membrane successfully. The incorporation of TiO$_2$ into the PAN polymer material endowed the UF membrane with a more hydrophilic, rough and porous surface. With the optimized fabrication condition of 15.0 wt% PAN and 0.6 wt% TiO$_2$, the M3 membrane presented the highest water permeation of 207.0 L m$^{-2}$ h$^{-1}$ bar$^{-1}$ and outstanding 99.0% BSA rejection. In addition, the incorporation of TiO$_2$ endowed the M3 membrane with a superior *FRR* of 37.13%, which was 2.28 times higher than that of the pristine M0 membrane, indicating that the PAN membrane decorated with TiO$_2$ enhanced the antifouling properties under water rinsing. Furthermore, remarkable self-cleaning ability was exhibited with the photocatalytic activity of TiO$_2$, and the value of *FRR* suffered a promotion to 52.41% and the BSA rejection remained constant. Consequently, the UF membrane incorporated with TiO$_2$ provided an effective method to conquer the membrane fouling problems in wastewater treatment.

**Author Contributions:** Conceptualization, J.L.; methodology, T.W.; resources, W.F.; data curation, X.W.; writing—original draft preparation, Y.X.; writing—review and editing, J.L.; visualization, H.L.; funding acquisition, W.F. All authors have read and agreed to the published version of the manuscript.

**Funding:** This research was funded by the Natural Science Foundation of Jiangsu Province, grant number: BK20211237.

**Data Availability Statement:** All data generated or analyzed during this study are included in this published article.

**Acknowledgments:** The financial support from the Natural Science Foundation of Jiangsu Province (Grant No. BK20211237).

**Conflicts of Interest:** There are no conflicting interest declared by the authors.

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
