# Peer review of "PAN/TiO2 Ultrafiltration Membrane for Enhanced BSA Removal and Antifouling Performance"

_catalysts, doi:10.3390/catal13101320_

Round 1
Reviewer 1 Report
The manuscript is well organized and the experimental work well carried out and interpreted. I only have serious concerns regarding the novelty as I think that the influence of the addition of TiO2 onto PAN membranes has been well investigated in previous reports. Even with TiO2 surface modifications which provided even better results, by considering the higher affinity of the nanoparticles with the matrix. This was not stated in the introduction and the literature was not properly cited and no comparisons was provided with the existing literature. I could find several papers on PAN modification by TiO2 with the aim of improving selected features. Authors must cite carefully the existing literature and evidence the novelty of the work with respect to the work reported. The only use of BSA cannot be a substantial reason of novelty. I hope that authors can highlight this aspect, otherwise the publication will be difficult. This is a major point for this work.
fair
Reviewer 2 Report
Membrane technology can be efficiently used for water and wastewater treatment purposes. But fouling reduces the capacity and separation efficiency of membrane processes and the life time of the membrane modules, respectively.
Detailed analysis of fouling behaviour, development of new cleaning processes and novel membrane materials can mitigate the reversible and/or irreversible fouling and prolong the life times of membranes, therefore the investment and operational costs can be reduced, as well. Photocatalytic materials using as casting agents in the fabrication of composite polymer membranes (such as PAN) can be a viable option.
Therefore, the topic of the manuscript catalysts-2601421 can be considered as relevant, the manuscript can provide interesting information for the readers.
SEM and XRD analysis of membrane structures, FTIR analysis of membrane surfaces, contact angle measurements for investigation of membrane-materials interactions (hydrophilicity, for instance), determination of (water)permeability and rejection, and membrane resistances analysis are accepted methods in membrane science. In lab-scale studies BSA is commonly used component to investigate the fouling behaviour in UF processes. Therefore, considering the specific aims of the research, the applied methods are adequate.
The manuscript is generally well structured. Introduction section summarizes well the background of the study based on relevant references.
Materials and methods are summarized clearly, but references and some data/information are missing from this section (see my comments, suggestions).
The manuscript contains interesting and novel results that are discussed with relevant references but need revision (see my specific comments).
Comments, suggestions:
[1.] Please give clearly the novelties of the study in the Introduction section, as well.
[2.] In my opinion, ANOVA and real multicriteria optimization of membrane fabrication conditions (mainly composition) are missing from the study.
[3.] Authors used BSA for testing the composite UF membranes. Please summarize briefly which types of wastewater can be modelled by BSA solution (scale-up potential).
[4.] Authors used a crossflow UDF module (line 308-309). Please give the details of the membrane unit.
[5.] Please check and correct the table and figure numbering. Section 3.1 (methodology) ‘Preparation of PAN/TiO2 membrane’ refers to Table 9 and and Figure 1, but the relevant data and information are summarized in Figure 9 and Table 2, respectively (see line 287-291), for instance.
[6.] Please make clear in the methodology section how was the TiO2 and Pan concentration range selected/determined.
[7.] Please give the meaning (and units) of W1 and W2 in Eq.1, as well.
[8.] Please give reference(s) for Eq.5-7.
[9.] The visibility if Figure 5 and Figure 6 is poor. please improve the quality of these figures.
[10.] Please use 0-100% scale for y axis of Figure 7.a (BSA rejection).
[11.] Please provide standard deviations of data presented in Table 1.
Round 2
Reviewer 1 Report
The paper is suitable for publication
english is ok
Reviewer 2 Report
The manuscript has an interesting, novel and relevant-for-practice topic. Authors have revised the manuscript thoroughly according to reviewers comments and suggestions and provided detailed answer for reviewers questions. Amendments, more detailed methodology and discussion part made the manuscript more complete and clear. The overall scientific quality of the MS has been improved significantly. I agree and accept all modifications made by the authors.